# Reciprocal Communication and Political Deliberation on Twitter

**Robert Ackland** [1] , **Felix Gumbert** [2] , **Ole Pütz** [2,*] , **Bryan Gertzel** [1] **and Matthias Orlikowski** [2]

[1] School of Sociology, Australian National University, Canberra 2601, Australia;
robert.ackland@anu.edu.au (R.A.); bryan.gertzel@anu.edu.au (B.G.)

[2] Faculty of Technology, Bielefeld University, 33615 Bielefeld, Germany;
felix.gumbert@uni-bielefeld.de (F.G.); matthias@orlikow.ski (M.O.)

* Correspondence: ole.puetz@uni-bielefeld.de

**Abstract:** Social media platforms such as Twitter/X are increasingly important for political communication but the empirical question as to whether such communication enhances democratic consensus building (the ideal of deliberative democracy) or instead contributes to societal polarisation via fostering of hate speech and "information disorders" such as echo chambers is worth exploring. Political deliberation involves reciprocal communication between users, but much of the recent research into politics on social media has focused on one-to-many communication, in particular the sharing and diffusion of information on Twitter via retweets. This paper presents a new approach to studying reciprocal political communication on Twitter, with a focus on extending network-analytic indicators of deliberation. We use the Twitter v2 API to collect a new dataset (#debatenight2020) of reciprocal communication on Twitter during the first debate of the 2020 US presidential election and show that a hashtag-based collection alone would have collected only 1% of the debate-related communication. Previous work into using social network analysis to measure deliberation has involved using discussion tree networks to quantify the extent of argumentation (maximum depth) and representation (maximum width); we extend these measures by explicitly incorporating reciprocal communication (via triad census) and the political partisanship of users (inferred via usage of partisan hashtags). Using these methods, we find evidence for reciprocal communication among partisan actors, but also point to a need for further research to understand what forms this communication takes.

**Keywords:** political communication; political deliberation; social network analysis; political partisanship; polarisation; Twitter; 2020 US presidential election

## 1. Introduction

Social media platforms give citizens the opportunity to gain information on political issues and actively participate in political communication. However, research suggests that social media may not bring us closer to the ideal of deliberative democracy (Habermas 1996, 2021), with concerns that social media use may contribute to societal fragmentation if it reinforces previously-held beliefs by limiting exposure to attitude-challenging information, due to user preferences to interact with like-minded people (homophilic selective exposure). Related concerns focus on social media as a potential driver of societal polarisation by fostering hate speech and the spread of mis/disinformation, thereby undermining the foundations of democratic consensus building.[1]

However, research into these phenomena usually does not account for a fundamental feature of social media; in addition to platform-specific forms of interaction such as likes or the sharing of content, there are also forms of interaction which support reciprocal communication between two or more users, such as comments or replies. Although reciprocal communication is at the core of deliberation and important for opinion-forming processes, it is not examined in studies that consider only the sharing of content or likes, such as the often-investigated retweet networks on Twitter/X (Barberá et al. 2015; Crupi et al. 2022; Gruzd and Roy 2014; Tyagi et al. 2020; Williams et al. 2015).[2] Furthermore,

in the studies that do consider replies (Arlt et al. 2018; de Franca et al. 2021; Yarchi et al. 2020; Yardi and Boyd 2010), reciprocity is only investigated in a limited sense as a two-step process of action and response. Such studies therefore do not account for the fact that reciprocity can involve action and response followed by further responses (on Twitter, the reply to an original tweet, followed by further replies to that reply). In summary, what is conventionally understood as a dialogue or discussion among two or more participants is not directly investigated in social media research on polarisation and fragmentation. To the best of our knowledge, we provide the first such investigation of sequentially organised reciprocal communication on Twitter.

In the first part of the paper, we describe the collection of a large-scale dataset of tweets related to the first debate of the 2020 US presidential election between Donald Trump and his contender Joe Biden in September 2020 (we refer to this as the "#debatenight2020" dataset). In the first part of the data collection we used a target set of debate- and election-related hashtags to identify relevant tweets, but we then used the v2 Twitter API to collect the whole "reply trees" (i.e., all replies to an original tweet, the replies to those replies, and so on) which Twitter calls "conversations", of which the hashtag-collected tweets are part of. Importantly, it was not a requirement that tweets collected as part of reply trees contain our target hashtags. We find that a hashtag-based collection alone would have only collected 1% of the debate-related Twitter activity. The resulting dataset maintains the reciprocity of communication for sequences of related replies (Gumbert et al. 2022).

In the second part of the paper, we present an examination of political deliberation on Twitter, extending the network-analytic approach of González-Bailón et al. (2010). Our analysis of deliberation takes place at two levels: reply trees and sequences of interactions which we call "reply chains" extracted from these reply trees (the reply chains can be represented as a network of users replying to each other). We construct the network measures of "prerequisites for deliberation" proposed by González-Bailón et al. (2010)—the maximum width of the reply tree (a proxy for representation) and maximum depth of the reply tree (a proxy for argumentation)—but we then extend these indicators of deliberation in two key ways.

First, just because a reply tree has a long root-to-leaf path (network depth) does not necessarily mean there is significant interaction taking place (at the extreme, the sequence could simply be one person replying to themselves). We use a triad census of the user network representation of the reply chains (focusing on the count of triads involving mutual ties) to provide more information on the argumentation dimension of deliberation. Second, we infer the political partisanship of Twitter users by ascertaining whether they use at least one of a manually-curated set of partisan hashtags relevant to the debate and the election. We then use political partisanship to provide nuance to our network measures of representation and argumentation; specifically, we investigate how partisan actors are involved in interactions that meet the prerequisites for deliberation and find that mixed-partisan interactions appear more argumentative and representative when we focus on reply chains instead of reply trees.

## 2. Data and Methods

We first collected all tweets containing at least one hashtag from a set of election- and debate-related hashtags, where the tweet was authored during the first US presidential debate held on 29 September 2020, including 15 min before and after the debate (Gertzel 2021a). This resulted in a dataset containing 2,387,587 tweets, 28,562 of which were reply tweets (authored by 26,869 users).

### 2.1. Collection of Reply Trees and Extraction of Reply Chains

Twitter defines "conversation threads" as consisting of an original tweet, the replies to that original tweet, and the replies to those replies (and so on). This activity can be represented as a tree structure that is instantiated when users use the reply function to respond to an original tweet, a tree that grows with every additional reply. The original

tweet acts as the root node of this tree, and all replies as well as replies to replies are nodes on branches of this tree, all sharing one conversation ID (the ID of the original conversation-starter tweet). We used the Twitter API v2 via the VOSON Lab R package voson.tcn (Gertzel 2021b) to identify the conversation ID for each of the 28,562 reply tweets (we focused on replies as they are the "building blocks" of reciprocal communication on Twitter, in contrast to retweets which are mainly used for information diffusion), and we then collected all reply tweets for each conversation ID. It is important to note that we collect the full reply tree for each conversation ID, i.e., the replies are included even if they do not contain debate-related hashtags. The data collection resulted in a dataset of 13,119 reply trees with a "conversation starter" tweet as the root node and all the subsequent replies and replies-to-replies (Figure 1(left)).[3] These reply trees contain 2,618,664 reply tweets (authored by 856,566 users) so the initial hashtag-based collection therefore missed 99% of the debate- and election-related reply activity and 97% of the users engaged in this activity.

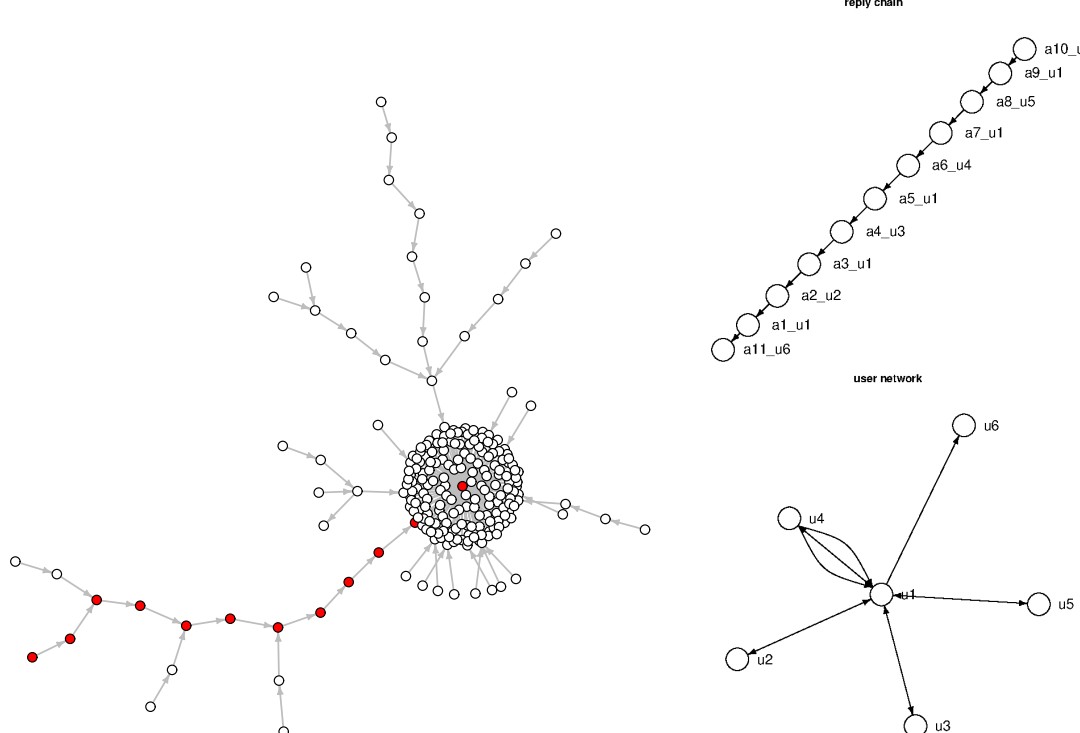

**Figure 1.** Reply tree and reply chain in red (**left**), and a reply chain and corresponding user network (**right**).

We extracted 2,119,655 reply chains from the 13,119 reply trees. A reply chain is a root-to-leaf path or branch: all the reply tweets from the root node (conversation starter) to a leaf node (the last reply on that branch); see the sequence of nodes in Figure 1(left). Each reply chain can be represented either as a network of reply tweets or as a network of the users who authored the reply tweets (Figure 1(right)). Figure 2 presents summary information on the length of reply chains in terms of replies and *turns* (sequential unique actors); the chain of replies authored by three users A→B→B→C→A has five replies and four turns. It is apparent that the vast majority of reply chains are very short: 86% of the chains contain only two replies and 96% of the chains contain two or three turns.

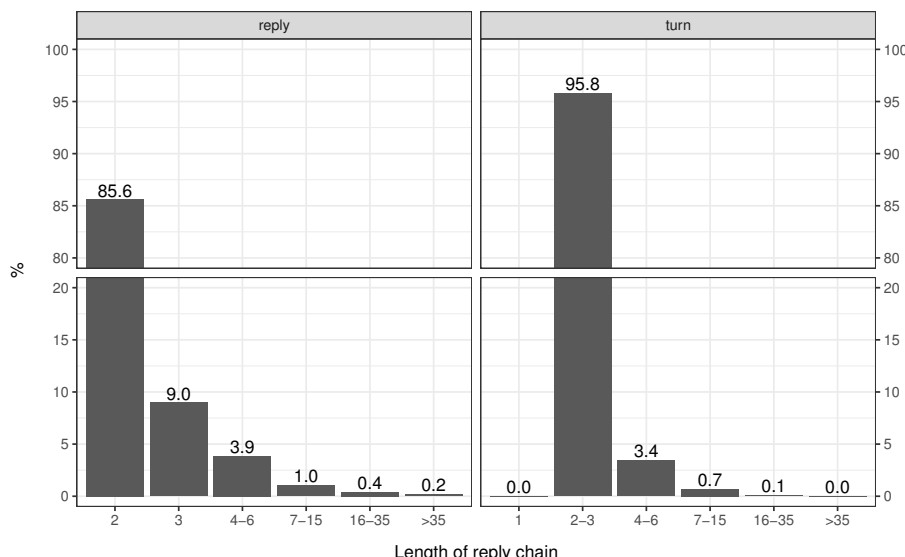

**Figure 2.** Composition of reply chains, based on number of replies (**left**) and number of turns (**right**).

### 2.2. Inferring Political Partisanship

Three (scalable) approaches for identifying partisan actors on Twitter have been proposed in the literature:

1.  Partisan clusters in retweet network: Conover et al. (2011) used modularity clustering and found that retweet networks (compared with, e.g., mentions networks) were more likely to exhibit clusters that were homogeneous in terms of composition of partisan actors.
2.  Usage of partisan hashtags: in their analysis of Twitter data for the first debate of the 2016 US presidential election, Rizoiu et al. (2018) labelled users depending on whether they had used one or more of a manually curated set of partisan hashtags.
3.  Partisan clusters in follows network: Barberá et al. (2015) constructed a follows network consisting of ordinary Twitter users and the partisan actors (e.g., politicians, media actors, think tanks) that they follow, and they then used correspondence analysis to identify partisan clusters.[4]

Given the significant challenge of collecting follows data for a dataset the size of #debatenight2020, we decided that identifying partisan clusters in follows network was not feasible. Given our focus is on reply ties (not retweets) and the vast majority of ties in our dataset are the former, we decided against using the partisan clusters in the retweet network approach. Thus, for this paper, we identified partisan actors using partisan hashtags. We first extracted 31,016 unique hashtags from the initial hashtag-based collection. We manually coded the 487 hashtags that have been used 50 or more times and identified 98 partisan left hashtags (examples: #votehimout, #dumptrump, #voteblue) and 23 partisan right hashtags (examples: #maga, #trumplandslide, #sleepyjoebiden). We then identified the partisan users among our 856,566 users present in the 13,119 reply trees, identifying 39,052 partisan actors (left: 32,584, right: 6468).

## 3. Results

First, we undertake an analysis of deliberation at two levels (reply tree, reply chain) and we then incorporate political partisanship into the analysis of political deliberation.

### 3.1. Two Levels of Analysis of Deliberation

In this section, we illustrate how to conduct analysis into deliberation at two levels: reply trees and the reply chains that can be extracted from these trees (and we focus on the user network representation of these reply chains).

### 3.1.1. Reply Trees

González-Bailón et al. (2010) highlight the fact that not all discussions count as deliberative—certain conditions need to be met—but these conditions are often normative and not conceptually well-suited for quantitative empirical application using large datasets that are available from social media. González-Bailón et al. (2010) draw on Ackerman and Fishkin (2002) who emphasise two "prerequisites of deliberation": *representation* (maximum representation means more people are involved in discussion increasing the diversity of viewpoints) and *argumentation* (maximum argumentation means there is more engagement and persuasion, leading to formation of preferences and opinion).

González-Bailón et al. (2010) operationalise these two prerequisites of deliberation by representing Slashdot discussion threads as radial tree networks where replies to the original post are in the first layer, replies to these replies in the second layer, and so forth. The number of comments at any layer approximates the number of different people involved in discussion and so the authors proxy representation by the maximum number of comments at any layer (maximum width). The number of layers through which discussion unfolds approximates intensity of argumentation (deeper trees indicate longer exchanges between participants) and so the authors proxy argumentation by the maximum depth of the network. Figure 3 illustrates the four zones of deliberation for reply trees (denoted using the numerals I–IV) using exemplar reply trees based on data in the present paper.

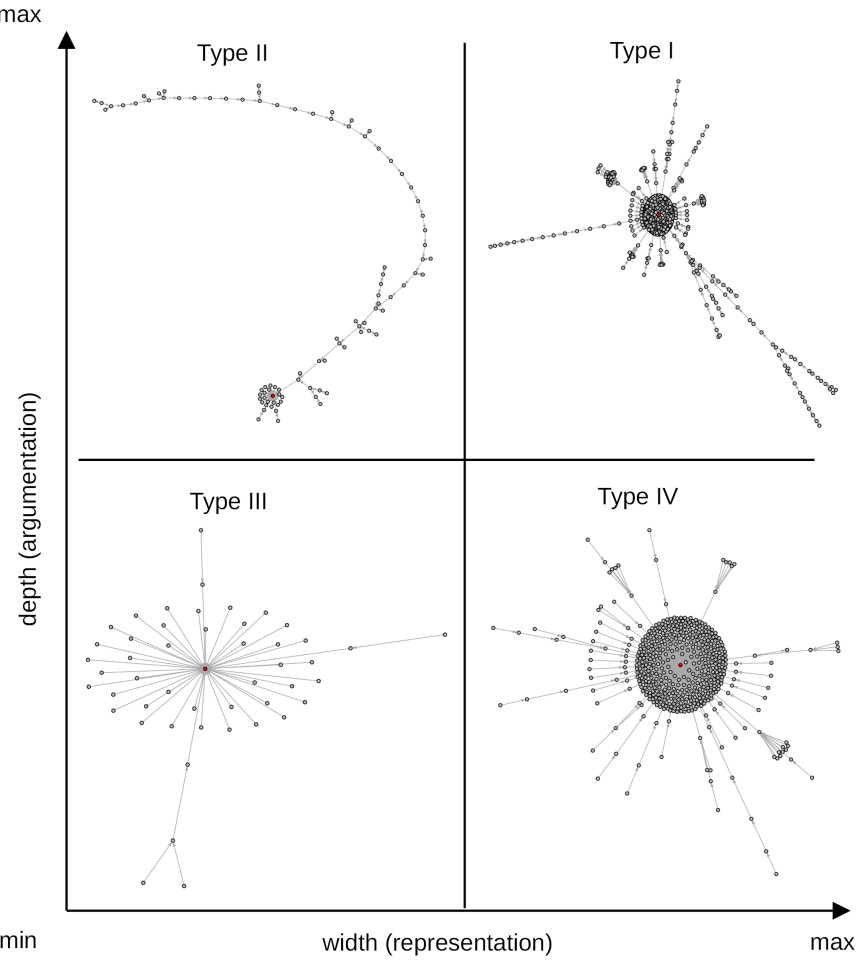

**Figure 3.** Network measures of deliberation using reply trees. The four zones of deliberation (based on maximum width and depth in the reply tree) are denoted using the numerals I–IV.

Figure 4 shows maximum width vs. maximum depth for a subset of 610 reply trees in the #debatenight2020 data, where depth $\geq 10$, $10 \leq$ width $\leq 2000$. The dotted lines divide the four zones according to the mean values for maximal width and depth of the

reply trees, which are 421.1 and 17.1, respectively. There were 11.3% of reply trees in Zone I (above mean in both width and depth), 15.7% in Zone II (below mean in width but above mean in depth), 50% in Zone III (below mean in both width and depth), and 23% of reply trees in Zone IV (above mean in width but below mean in depth). This suggests that there are more reply trees with a higher than average level of representation (Zone IV), than argumentation (Zone II). The lowest proportion of reply trees is located in Zone I, which is characterised by argumentation as well as representation. Overall, the plot clearly shows the diversity of the 610 reply trees.

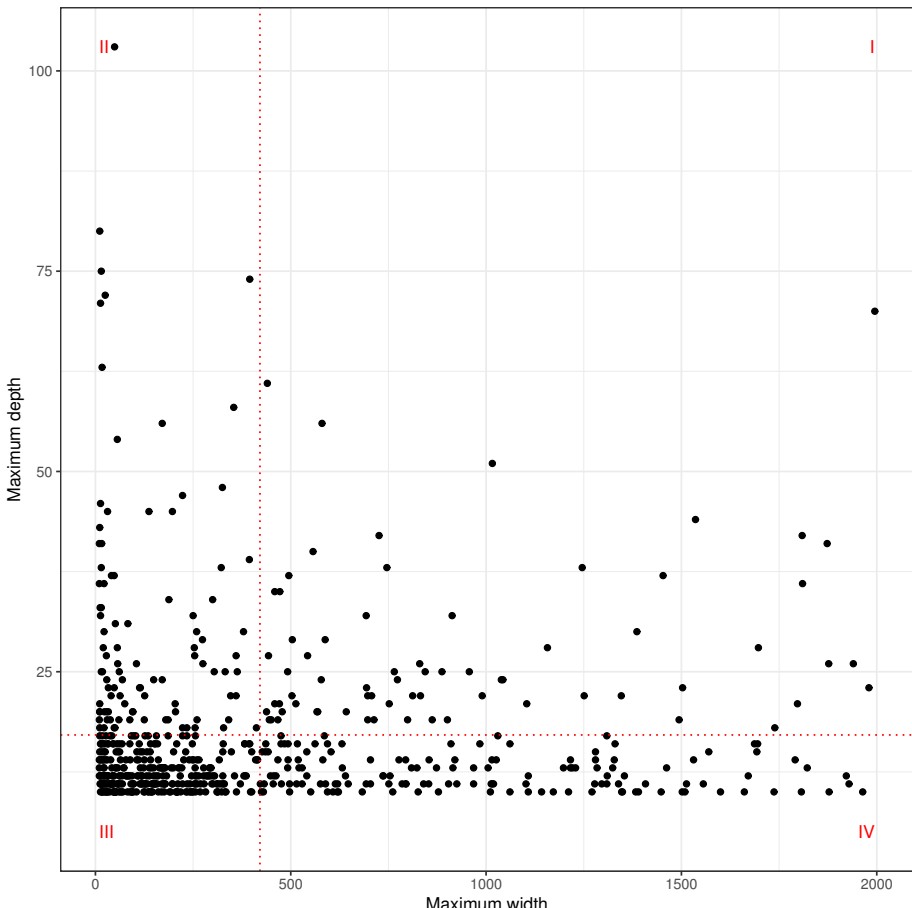

**Figure 4.** Maximum width and depth of reply trees. The dotted red lines indicate means of maximum width and depth, and demarcate the four (reply tree) deliberation zones.

### 3.1.2. Reply Chains

The reply trees that are located in Zone I in Figure 4 have relatively high maximum depth (as well as high maximum width), but that does not necessarily mean that argumentation is occurring across the trees in this Zone. For example, there may be one long root-to-leaf branch in a reply tree, while the other branches are short. Such a tree would nonetheless be found in Zone I, even though the distribution implies that only few users engage in argumentation, while there is little reciprocal engagement from the majority of users. Indeed, Figure 2 shows that the vast majority of root-to-leaf branches, which we call reply chains, are short. Furthermore, it is unlikely that the participants in one branch of the tree are always aware of interactions in other branches of the tree, so it can be misleading to treat all users who are active in a reply tree as taking part in one giant "conversation" (although this is how Twitter describes reply trees). We therefore investigate interactions further by undertaking a triad census (Davis and Leinhardt 1972) of the user network representations (where nodes are users) of the reply chains from the subset of 610 reply trees. In a triad census, every triple of vertices (A, B, C) are classified into 16 possible states. For example, the configuration

[A→B, C], the graph with a single directed edge, is state 012, [A↔B, C], the graph with a mutual connection between two vertices is state 102, and [A←B→C], the out-star, is state 021D. Note that we exclude from consideration reply chains that are only composed of the original (conversation starter) tweet plus a reply. We exclude such very short reply chains because by definition, they cannot represent reciprocal communication and cannot be considered to approximate argumentation. We further exclude all original tweets from consideration and only focus on the replies in chains. The reason for this is that most hashtags are used in original tweets, which affects the classification of a chain as partisan, but many original tweet authors in our dataset are public figures that rarely engage in reciprocal communication; the content of their tweet serves as a topic for discussions of other users. In short, by excluding very short reply chains and original tweets, we focus more narrowly on partisan interactions in reply chains as a prerequisite of deliberation. Figure 5 shows user networks (and their triad census) constructed from exemplar reply chains from each of the four zones of deliberation, which we denote with the numerals 1–4.

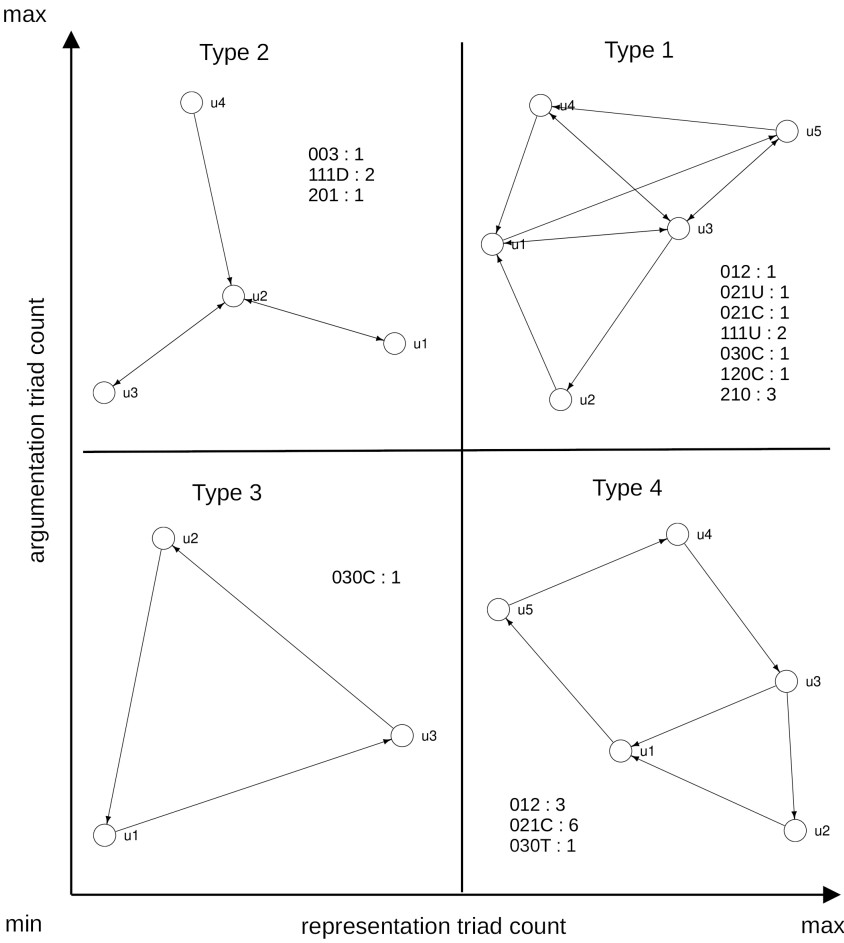

**Figure 5.** Network measures of deliberation using reply chains. The four zones of deliberation (based on representation and argumentation triad count) are denoted using the numerals 1–4.

Excluding the null triad state (no edges between the three nodes) we allocated the 15 other triad states into two groups: *representation triads* do not contain mutual ties while *argumentation triads* do contain mutual ties. Similar to the definition of representation and argumentation for reply trees above, representation triad counts for reply chains approximate the extent to which multiple users participate, whereas argumentation triad counts approximate the intensity of reciprocal exchanges between those users. Of the 314,550 reply chains extracted from the 610 reply trees, a subset of 20,777 chains had

non-zero counts of either representation or argumentation triads. This subset of chains is plotted in Figure 6; the dotted lines are drawn at the mean values for the representation and argumentation triad counts (1.5 and 0.9, respectively) and indicate boundaries to the four zones of deliberation for reply chains.[5] There were 11.4% of the chains in Zone 1 (above mean in the counts of both representation and argumentation triads), while 32.2% were in Zone 2 (below the mean in count of representation triads but above the mean in argumentation triad count), 46.4% of reply chains were in Zone 3 (below the mean in the counts of both representation and argumentation triads), and 10% were in Zone 4 (above the mean in count of representation triads but below the mean in argumentation triad count).

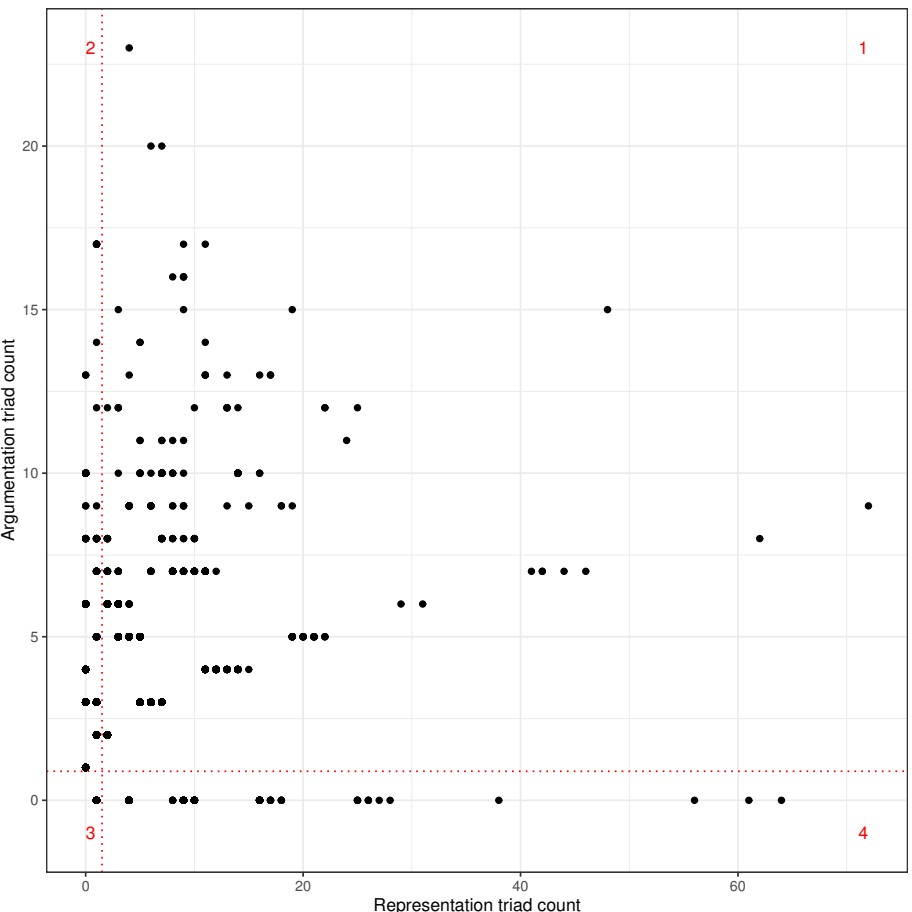

**Figure 6.** Counts of representation and argumentation triads in reply chains. The dotted red lines indicate means of triad counts, and demarcate the four (reply chain) deliberation zones.

### 3.2. Incorporating Political Partisanship into Measures of Deliberation

In this section, we examine how the incorporation of partisanship into the analysis changes our insights into deliberation occurring on Twitter during the debate.

#### 3.2.1. Reply Trees

We created a partisanship score for each of the 610 reply trees in Figure 4, which is the ratio of the proportion of partisan actors in the reply tree to the average of this proportion across all 610 trees. We labelled the 305 trees with partisan scores in the top 50% as "partisan" and we further classified the partisan reply trees as "partisan-right" when left actors accounted for less than 30% of partisan actors in the tree, "partisan-mixed" when left actors account for between 30 and 70% of all partisan actors in the tree, and "partisan-left" when left actors account for more than 70% of partisan actors in the tree. Of the 305 partisan reply trees, 262 were left, 35 mixed and eight right. Figure 7(left panel)

shows the maximum width and depth of the partisan reply trees (colour-coded according to whether partisan-left/right/mixed) and the right panel shows the distribution of reply trees across the four zones of deliberation. Zone I contains 11.3% of all 610 reply trees and 14.1% of the 305 partisan-left trees, 5.7% of the partisan-mixed trees, but no partisan-right trees.

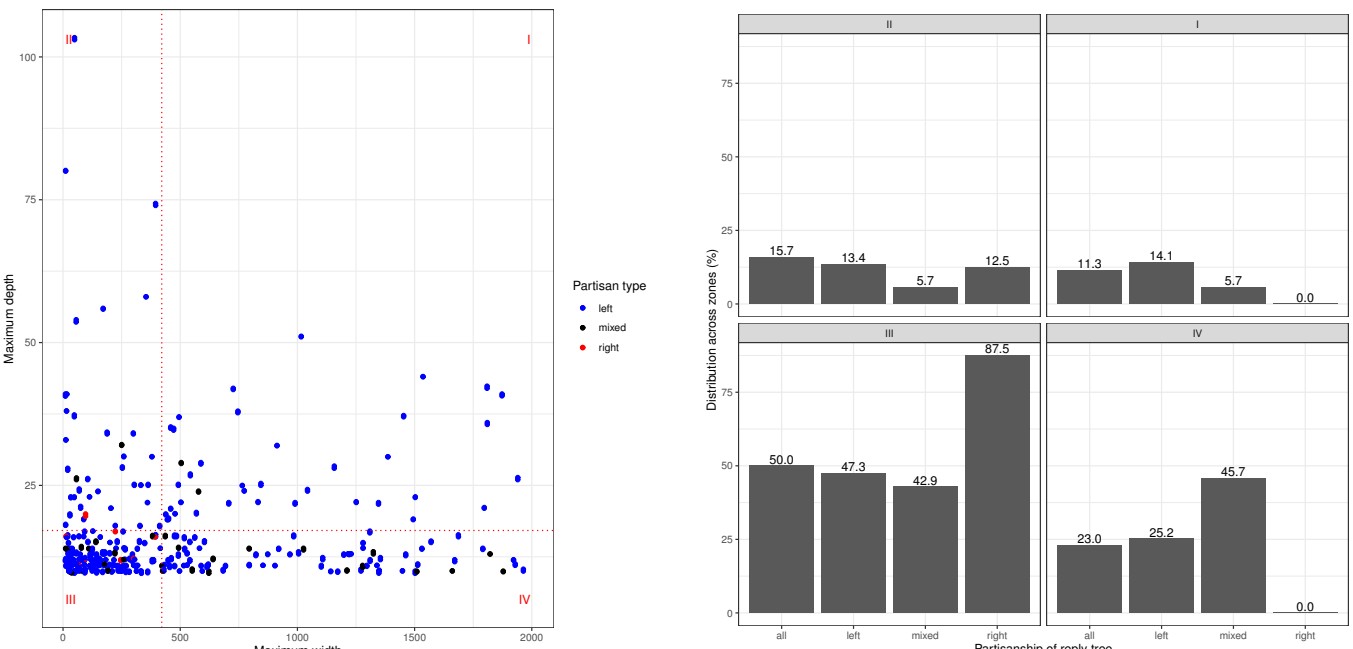

**Figure 7.** Maximum depth and width of partisan reply trees (**left**) and the distribution of partisan and all trees across the deliberation zones (**right**).

The partisan coding of Twitter users leads to further insights into deliberation across the partisan divide and also among partisan actors of the same political group; in Figure 7, we find partisan-mixed trees represented in all zones. The largest percentage of mixed-partisan trees (45.7%) can be found in Zone IV, which is characterised by above mean representation and below average argumentation. This suggests that when there is interaction across the partisan divide, a larger number of users is involved, but these users appear to engage in comparatively short reciprocal exchanges. Partisan-left reply trees can be found in all four zones as well, closely reflecting the overall distribution of trees in all zones. Reply trees with a majority of partisan-right actors, on the other hand, can only be found in Zone III, which is characterised by below mean argumentation and representation, and Zone II, characterised by above mean argumentation and below mean representation.

### 3.2.2. Reply Chains

We used the same approach to identify partisan reply chains (although we used a 10% threshold for the partisan score, instead of 50%) and 880 of the 20,777 chains were classified as partisan left/right/mixed. Of the 880 partisan reply chains, 697 were left, 82 mixed and 101 right. Comparing partisan with all reply chains, while we find a similar maximum of the count of argumentation triads, there is a much smaller maximum of the count of representation triads in partisan reply chains (compare x-axis in Figures 6 and 8).

In Figure 8, we find left/right/mixed reply chains in all four zones.[6] Partisan-mixed triad chains are over-represented in Zone 1, where we find chains with a count above mean of both argumentation and representation triads. Both partisan-left and partisan-right reply chains are over-represented in Zone 3, which is characterised by below mean counts of both argumentation and representation triads.

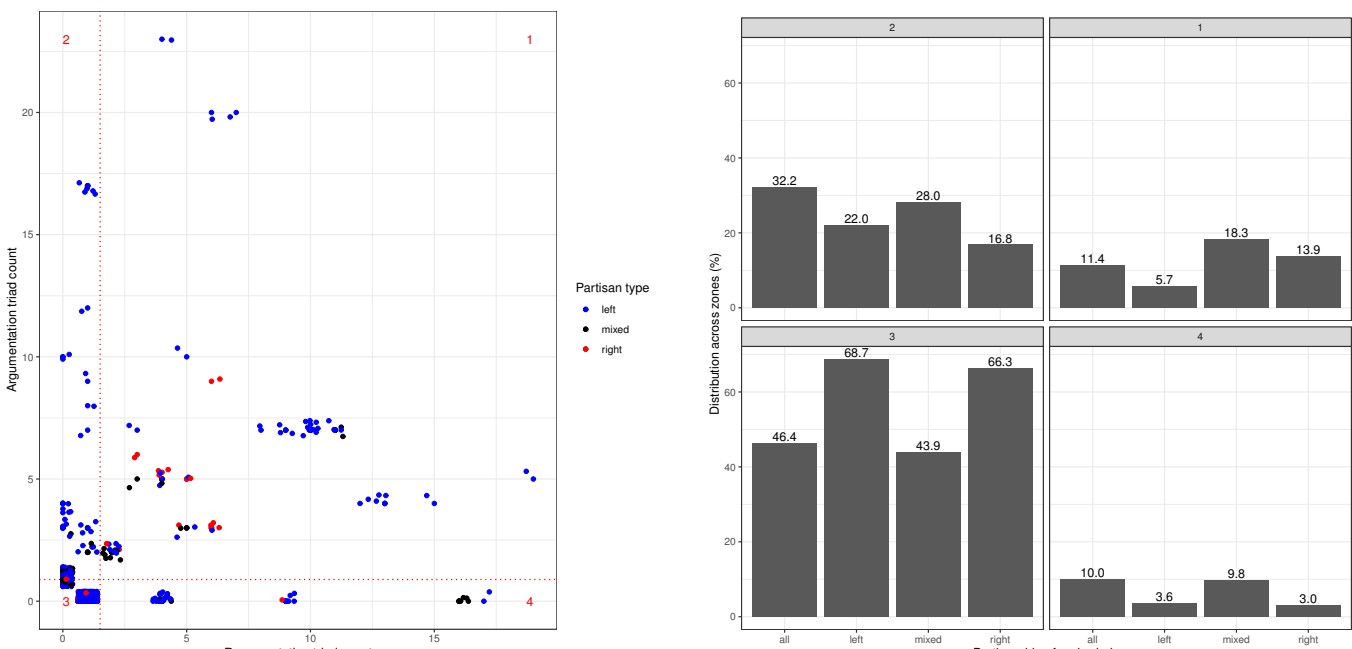

**Figure 8.** Counts of representation and argumentation triads in partisan reply chains (**left**) and the distribution of partisan and all chains across the deliberation zones (**right**).

## 4. Discussion

We argued that network analytic studies often describe political interactions on social media as fragmented by partisanship, but such studies do not investigate reciprocal communication as a key aspect of deliberation. We show that the investigation of reply trees during the first US presidential debate in 2020 does indicate that social media interactions meet "prerequisites of deliberation" (Ackerman and Fishkin 2002; González-Bailón et al. 2010). Representation and argumentation are approximated by tree depth and width, and the zones are divided by the mean values for maximal depth and length of trees. With this approach, we find comparatively more reply trees that are representative (Zone IV, 23%) than trees that are either argumentative (Zone II, 15.7%) or argumentative and representative (Zone I, 11.3%).

We further extended the foundational work of González-Bailón et al. (2010) in two ways: first, we introduced the use of triad census data to approximate representation and argumentation within reply chains, where chains are the branches of a reply tree. Second, we incorporated political partisanship into the measurement of deliberation through the use of partisan hashtags, allowing us to explore the occurrence and extent of cross-partisan reciprocal communication as well as communication within partisan groups.

With partisanship incorporated into the analysis of reply trees, we find that trees that are defined as partisan-mixed are predominantly located in Zone IV, characterised by below average argumentation and above average representation. This suggests that when there is interaction across the partisan divide, a larger than average number of users is involved, but these users appear to engage in comparatively short reciprocal exchanges. We should recall, however, that partisanship is defined by the ratio of the proportion of partisan actors in the reply tree to the average of this proportion in all trees, which means that there are also other users present who do not use partisan hashtags and whose political inclinations are unknown. Furthermore, the classification of a reply tree as partisan-left (or partisan-right) by the proportion of partisan-left (or partisan-right) actors involved can hide the fact that there may be cross-partisan interaction in its branches.

On the level of reply chains, where we approximate representation and argumentation using triad census counts, we find a somewhat different picture, with partisan-mixed reply chains being over-represented in Zone 1, while partisan-left and partisan-right reply

chains are both over-represented in Zone 3. This suggests that partisan-mixed interactions tend to be more reciprocal (argumentative) and involve more users (representative) than non-partisan reply chains. Interactions within partisan groups, on the other hand, tend to be less argumentative and representative, i.e., they involve fewer users and are less reciprocal. This might reflect that users with the same political affiliation share similar opinions and preferences and are thus less likely to engage in discussions where such opinions are challenged by other users with a different political stance.

Although the results from trees and chains cannot be directly compared due to different methods of approximating representation and argumentation, it is worth highlighting the following difference: overall, we find that mixed-partisan interactions appear more argumentative and representative when we focus on the level of reply chains instead of the level or reply trees. This suggest a need for a closer inspection of replies as forms of political communication, as we outline in the next section.

## 5. Conclusions

In this article, we argued that empirical research on political deliberation online should consider reciprocal communication between users as an important element of opinion-formation processes. This presupposes an approach of data collection that is able to reconstruct interactions between users in their sequential order represented in the form of reply trees and reply chains, where chains can be extracted from trees. With the #debatenight2020 dataset, we were able to increase the total number of tweets in our dataset by a factor of 2 from roughly 2.3 million tweets (hashtag-based collection) to nearly 5 million tweets (hashtag-based plus conversation ID-based collection). Furthermore, we were able to increase the number of replies by a factor of 92 from 28,562 to a total of 2,618,664.

This means that our initial hashtag-based collection only includes 1.1% of replies that were written by users as part of Twitter conversations pertaining to the first presidential TV debate between Donald Trump and his contender Joe Biden. Up to 98.9% of this kind of reciprocal communication would have been missed by a collection based on our target hashtags alone. This method of data collection is easier to implement than a reconstruction via mentions (Muhle et al. 2018) or the reply_to_tweet_id, which only links pairs of tweets (Moon et al. 2016; Scheffler 2017), and it provides a broader and more complete picture of reciprocal communication on Twitter, thus shedding light on the blind spots of existing research that focuses on hashtags alone. To the best of our knowledge, this is the first paper where large-scale Twitter political reply networks are constructed and used for network analysis of political deliberation. While Hada et al. (2023) collect politically-oriented Twitter interactions through replies, their analysis neither focuses on nor requires the preservation of the reciprocal nature of Twitter replies.

With the analysis of reply trees and reply chains, we are also introducing methods that have not received much attention to date in fragmentation and polarisation research. Since deliberation is conceptually antagonistic to fragmentation (users that do not encounter one another cannot deliberate), the claim that the public sphere is fragmented requires some qualification. As opposed to networks constructed via retweets, mentions, or follows, the reconstruction of reply trees and chains highlights that there is interaction via replies between partisan actors which meets prerequisites for deliberation: there are more partisan-mixed reply chains with above-mean counts of representation and argumentation triads than the overall distribution of reply chains would suggest. Nonetheless, this observation needs to be put into perspective by the fact that the vast majority of reply chains are rather short. This might indicate that while partisan users are exposed to a diversity of opinions, engagement with the other side is limited in absolute terms, which may rather serve to reinforce polarisation. Given these ambiguous results, further research into reply trees and chains is necessary.

Specifically, there are three limitations to the methods we employed that need to be addressed to advance our understanding of political deliberation online.

First, our approach for identifying partisan actors via usage of partisan hashtags will be subject to error because it will miss partisan actors who did not use one of our partisan hashtags. Overall, partisanship could only be identified for 4.5% of users via hashtags, although the Pew Research Center presented evidence that up to 36% of adult US-American Twitter users identify as democrat and 21% as republican (Wojcik and Hughes 2019). Furthermore, a qualitative inspection of a sample of reply chains suggests that hashtags are rarely used in longer reply chains, which is further evidence that the methods used in this paper may potentially underestimate partisan interactions.

Second, while we expand on the prerequisites of deliberation proposed by González-Bailón et al. (2010) by applying the triad census method to user network representations of the reply chains, we note that this method does not preserve the timestamps on edges. This can be illustrated with the example of the Type 3 network in Figure 5, where the single triad state of 030C does not provide information on the sequential order in which the replies occurred. We suggest that methods such as network motifs (Lehmann 2019) may be better suited for the dynamic analysis of reply chains, but they also require a more differentiated classification of replies.

Third, while the measures of prerequisites for deliberation used in this paper are useful because they can be effectively scaled, their use naturally leads to the question of which proportion of cases truly meets the criteria of deliberation. This particularly concerns the question whether the length of both reply trees and chains is a good approximation of argumentation. For example, among the set of all reply chains, only a subset may actually contain sustained conflictual interactions where differing viewpoints are expressed, while another subset may contain non-conflictual interactions that only serve to reinforce existing viewpoints. While out of scope for the present paper, this question can only be answered based on detailed qualitative analysis. Such qualitative analysis, however, is not possible to perform at scale and hence would be restricted to a smaller sample. A potential solution would be to use automated text analysis using Large Language Models for social media text (Antypas et al. 2023; Nguyen et al. 2020), fine-tuned to replicate the categorisations found in the qualitative analysis. The use of such an approach will allow an exploration of whether the classification of reply chains as conflictual or non-conflictual can complement existing methods for approximating political partisanship.

**Author Contributions:** Conceptualization, All; methodology, All; software, B.G. and R.A.; validation, R.A.; formal analysis, All; investigation, All; resources, R.A.; data curation, R.A.; writing—original draft preparation, R.A., F.G., O.P. and M.O.; writing—review and editing, R.A., F.G., O.P. and M.O.; visualization, R.A.; supervision, R.A. and O.P.; project administration, R.A. and O.P.; funding acquisition, O.P. and R.A. All authors have read and agreed to the published version of the manuscript.

**Funding:** This research is part of the "Bots Building Bridges (3B): Theoretical, Empirical, and Technological Foundations for Systems that Monitor and Support Political Deliberation Online" project that was funded by the Volkswagen Foundation (AI and Society of the Future Stream).

**Institutional Review Board Statement:** The study was conducted in accordance with approval by the Human Research Ethics Committee of The Australian National University (protocol code H/2023/1228, approved 20 October 2023).

**Informed Consent Statement:** Not applicable (nor feasible) given this study involves a large-scale dataset collected via the Twitter API.

**Data Availability Statement:** The data from this study are not available to other researchers due to restrictions on the use of the Twitter API. Relatedly, our current ethics approval does not cover release of derived datasets.

**Conflicts of Interest:** The authors declare no conflict of interest. The funders had no role in the design of the study; in the collection, analyses, or interpretation of data; in the writing of the manuscript; or in the decision to publish the results.

## Notes

[1] See Arora et al. (2022); Iandoli et al. (2021); Lorenz-Spreen et al. (2023); Ludwig and Müller (2022); Terren and Borge-Bravo (2021) for systematic reviews. For hate speech in particular, see Castaño-Pulgarín et al. (2021); Strippel et al. (2023); Tontodimamma et al. (2021).

[2] Findings on polarisation and fragmentation are dependent on the form of interaction that is used to construct networks. Results suggest that retweet networks tend to show a higher degree of homophily than mention networks (Conover et al. 2011; Valle and Bravo 2018) or reply networks (de Franca et al. 2021). Twitter was rebranded as X in July 2023, but we use the former name in this paper.

[3] The data collection was completed by January 2022 (16 months after the debate), and the significant period of time between the debate and the completion of data collection would have resulted in loss of data due to user- or Twitter-initiated deletion of user accounts or tweets.

[4] This approach was used by Ackland et al. (2019) in their analysis of differences between the right and left in terms of diffusion of news via retweets.

[5] Note that the boundaries of the zones are close to zero because there are many short reply chains.

[6] Note that in Figure 8(left) the points are jittered to adjust for significant over-plotting.

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
