# Peer review of "Reciprocal Communication and Political Deliberation on Twitter"

_socsci, doi:10.3390/socsci13010005_

Round 1

Reviewer 1 Report

Comments and Suggestions for Authors

The manuscript under review presents an intriguing exploration of political discussions on digital platforms, focusing on Twitter as a case study. The study underscores the vital role of political discourse in shaping public opinion within the realm of social media. The research is both timely and relevant, given the increasing influence of digital platforms on political narratives. The study's methodology and approach are commendable, shedding light on an essential aspect of contemporary communication.

One significant strength of the research lies in its emphasis on the importance of political discussion on platforms like Twitter. The authors have adeptly captured the essence of the digital discourse, highlighting its implications on public perception and political engagement. However, to augment the depth of the analysis, it is essential to delve further into the textual content of the collected sample.

I recommend incorporating a text analysis component, specifically a frequency analysis of the most commonly used words in the dataset. This analysis would enrich the study by providing a nuanced understanding of the predominant topics discussed by Twitter users. By identifying the frequently used words, the authors can offer readers a more insightful view of the content and themes prevalent in political discussions on the platform.

Additionally, I suggest the inclusion of visual aids to enhance the presentation of the data. Specifically, the manuscript would benefit from the incorporation of two bar plots illustrating the top 20 most frequently used words among partisan groups, specifically the 'red' and 'blue' factions. These visual representations would offer a clear and concise overview of the distinctive vocabulary employed by different political affiliations, thereby enhancing the reader's comprehension of the findings.

In summary, while the manuscript is compelling and addresses a crucial aspect of digital communication, there is room for improvement in terms of textual analysis and data visualization. The inclusion of a frequency analysis of commonly used words and the integration of bar plots for partisan groups would significantly enhance the manuscript's overall quality, providing readers with a more profound insight into the nuances of political discussions on Twitter. I strongly recommend revisiting the analysis to incorporate these suggestions, which would undoubtedly strengthen the research and contribute to its scholarly impact.

Author Response

We thank the reviewer for the useful comments on our manuscript.

We agree that an analysis of the text content of the tweets in our dataset would be very interesting and we do have plans to undertake such work in the future. However, we feel that this is outside the scope of the present paper, which focuses on network analysis of political deliberation. The manuscript was already quite long and complicated. We have tried to simplify the manuscript in the revision, and we feel that adding text analysis would have made this process simplifying more challenging, and would have detracted from the network analysis.

Reviewer 2 Report

Comments and Suggestions for Authors

Overall, the paper includes interesting results. However, in my opinion it needs several modifications to improve its quality (and be considered for publication in this journal).

1) The paper lacks an adequate bibliography, which is quite limited in terms of similar studies on Twitter and other social media networks, as well as the background context on political communication. In summary, I found the theoretical framework to be significantly weak.

2) The authors should consider including a section that introduces research questions or hypotheses along with relevant content/descriptions. Adding this section would benefit the paper and provide readers with a clearer understanding of the study's scope.

3) I couldn't find any information regarding the time frame for data gathering. It would be helpful to know how long the data collection process lasted.

4) Finally, I believe the paper would benefit from an additional section summarizing the study's conclusions. While some points are mentioned in the Discussion section, it might be more effective to include them in a separate section, perhaps along with discussions of future directions for research.

I hope my comments will help the author(s) improve this paper.

Author Response

We thank the reviewer for the useful comments on our manuscript.

Our responses to the four comments are:

(1) We agree that the theory and literature review were under-developed and we have improved this aspect of the manuscript. In terms of theory, we feel the manuscript is now better framed (in the introduction section) in terms of what we mean by "reciprocal communication" and how this is crucial for deliberation (we emphasise that there cannot be deliberation without reciprocal communication). Furthermore, we explain how studies that focus on mentions, retweets, and likes do not investigate reciprocity.

(2) We agree that research questions/hypotheses could have been useful, but given this is exploratory research, we have opted to not include research questions. However, in our re-drafting of the paper we have endeavoured make the presentation clearer by emphasising the two levels of analysis (reply trees and reply chains) and we have removed material on the distribution of users (influencers, hidden influencers etc) which was not adding much to the analysis.

(3) Endnote 3 now provides information on how long the data collection took.

(4) We have included a conclusion section where we summarise some of the limitations of the study and our plans for future research.

Reviewer 3 Report

Comments and Suggestions for Authors

The authors present a paper titled "Reciprocal Communication and Political Deliberation on Twitter." The paper focuses on new research methods to delve deeper into political deliberation on social media, specifically on Twitter. The authors concentrate on the discourse surrounding the 2020 US general elections within the conversations on this platform.

The authors employ a methodology that enables the study of political deliberation, with a focus on the analysis of reply trees, reply chains, and users within Twitter conversations, along with the innovative inclusion of partisanship for the study of political deliberation. The article is well-written, clear, and structured appropriately. The results have been adequately presented and cover a highly relevant topic in today's context.

The following recommendations are provided to enhance the comprehensibility of the methodology and results:

-        Between lines 124 and 129, in section 2.2, when explaining the partisanship selection method, there is a significant difference in the number of hashtags selected for both sides. Could this affect the findings explained in section 3.2? What is the reason for this difference? It is recommended to provide a more thorough explanation of the consequences of this difference and its potential implications.

-        Starting from line 158, the authors refer to zones I, II, etc. It is assumed that this refers to the types found in Figure 3. Clarity is requested to facilitate understanding.

-        In line 163, the term "triad census" is used, as also seen previously in line 65. A somewhat more detailed, albeit concise, explanation of this concept can aid in making the text more accessible, particularly regarding the interpretability of the results.

-        Clarification is requested regarding the cutoff points that determine the depth of argumentation and breadth of representation in the results (Figures 4, 6, 8, ...). Are these determined through quantiles? Are they theoretical cutoff points?

-        In lines 278 and 281 of section 4, the authors mention contradictory results. Could this be due to the methodology used to define partisanship?

-        Additionally, it is recommended that the conclusions section presents a discourse that more clearly establishes the relationship between the applied methodology and the theoretical discussion of the results. This will help readers better understand how the methodology used in the study directly contributes to the insights and implications drawn from the findings, enhancing the overall coherence and impact of the paper.

Author Response

We thank the reviewer for the useful comments on our manuscript.

Regarding the difference in the numbers of left/right partisan hashtags identified, we believe that this difference reflects the difference in the underlying population: there are (or were) more democrat supporters on Twitter compared with republican supporters. We discuss this as the first limitation of the study, in the (now included) conclusion section. We note that we are under-estimating the number of partisan actors and further, we believe that the partisan hashtag approach is relatively worse at identifying partisan-right actors, compared with partisan-left actors. This is something we hope to address in future work.

Regarding the use of Zones and Types, we have tried to make this easier to follow. One thing that we have done is to remove material on the distribution of users (influencers, hidden influencers etc) as this seemed to be complicating the manuscript and didn’t add much to the analysis. But we have also tightened up the use of the Zone/Type terminology including using numerals I-IV for reply trees and numerals 1-4 for reply chains. We also include a revised conceptual figure for reply trees (Figure 3) and the corresponding figure for reply chains (Figure 5).

Regarding the use of triad census counts for reply chains, we have attempted to make this clearer by including a new figure (Figure 5) which shows four exemplar reply chains (or user network representations thereof) for each of the four zones of deliberation.

We have clarified that the boundaries for the zones of deliberation (Figures 4 and 6) are simply the means of the max width or depth for reply trees (Figure 4) or the representation or argumentation triad counts for reply chains (Figure 6). Gonzalez-Bailon et al. (2010) also used the mean (for width/depth). These boundaries are not theoretically determined, and the use of other cutoffs could lead to qualitatively different results.

In this revised manuscript we have attempted to better summarise our findings and we note in the conclusion that there is some ambiguity that could be resolved via addressing three limitations with our approach.

Finally, we agree that a conclusion section was required, and have now added one.

Round 2

Reviewer 2 Report

Comments and Suggestions for Authors

The author(s) have addressed the points raised in my previous review and I am satisfied that they have met the requested revisions. The manuscript has been significantly improved after the revision. It is a topical study with interesting results. I have no further worries about this paper and I support and recommend publication.